# OpenReview forum: "Eliciting Harmful Capabilities by Fine-Tuning on Safeguarded Outputs"
_ICLR.cc/2026/Conference — ICLR 2026 Poster_

### Official Review · Reviewer_d887 · 2025-10-15

**Soundness:** 3
**Presentation:** 3
**Contribution:** 3
**Rating:** 4
**Confidence:** 3

**Summary:**

This paper investigates elicitation attacks, a novel method of circumventing safeguards on frontier models. Instead of directly extracting harmful knowledge, the authors use frontier models to generate harmless but adjacent outputs, and then fine-tune open-source models on these outputs. The experiments, conducted primarily on chemical synthesis and processing tasks, show that fine-tuned open-source models regain up to ~40% of the performance gap relative to unguarded frontier models.

**Strengths:**

1. The paper identifies a less explored but realistic pathway for adversaries: leveraging benign outputs to indirectly reconstruct harmful capabilities.
2. The authors evaluate the proposed method across multiple open-source models.

**Weaknesses:**

1. The work convincingly shows uplift in chemical synthesis, but it is not entirely clear how an adversary would scale this approach to more complex or diverse malicious domains in practice, such as cybersecurity, disinformation, and bioterrorism. The paper would benefit from a stronger justification that such attacks are not just proof-of-concept but present a practical risk.

2. The paper argues that rubric evaluation based on keyword matching is an unreliable measure, and introduces structured comparison evaluation as a more fine-grained alternative. However, this method still primarily measures similarity to reference answers rather than the ability to successfully complete harmful tasks. As a result, the evaluation outcomes are heavily influenced by the quality and biases of the frontier models used to generate those references. Consequently, it remains unclear how much of the reported “40% performance gap recovery” actually translates into genuine security threats.

**Questions:**

None

---

> ### Author Response · Authors · 2025-11-27
>
> Thank you for recognizing that our work identifies a less explored but realistic attack pathway and for appreciating our evaluation across multiple open-source models.
>
> Addressing your specific points:
> 1. **“...not entirely clear how an adversary would scale this approach to more complex or diverse malicious domains in practice, such as cybersecurity, disinformation, and bioterrorism…”** We agree that it is unclear how to adapt this attack to all domains. We focus on scientific domains, and chemical weapons in particular, as they represent potentially very dangerous threats, and are the explicit targets of several frontier labs’ safety policies. We discuss this in Section 3.1. We believe that this attack could be adapted for bioterrorism by training on benign biological tasks as well.
> 2. **“...this method still primarily measures similarity to reference…”** We have clarified in the text (see top-level comment) that this is not what the evaluation is meant to do. The judge model in fact is not even told which of the two responses in the input is the anchor and which is the tested output. The role of the anchor / reference in the evaluation is to provide a calibration point so that the judge model outputs scores in a consistent manner, rather than varying wildly between contexts. This obviates the need for defining explicitly how response quality corresponds to the scoring scale, which would require a ground truth correct response.
> 3. **“...evaluation outcomes are heavily influenced by the quality and biases of the frontier models used to generate those references…”**  We have several lines of evidence that indicate this is likely not an issue:
> - We explicitly analyze how the quality of the references / anchor responses affects how accurately the evaluation captures response quality. We find in Section 3.4.3 that ground truth solutions are rated highly, even though the anchors are of questionable quality. This indicates that low quality references do not hurt the evaluation very much.
> - In Appendix E.1, we analyze how consistent evaluator transcripts are and find that generally, the same claims are often made by the judge model when resampling the evaluation. While the results in this section suggest that higher quality anchor responses lead to more consistent evaluations, the effect size is small.
> - We find in Section 3.4.1 that despite any biases in the evaluation procedure human experts largely agree with them. In particular, anchored comparison transcripts are rated by the experts as high quality, and the preferences of the anchored comparison align with that of the human experts.
> - We explicitly test several desirable properties of our anchored comparison throughout section 3.4 and find that it does better than evaluations from the literature. We find that anchored comparison evaluations are good at finding mistakes in responses, compared to other evaluations from the literature (Section 3.4.2). We find that anchored comparisons rate ground truth responses highly, whereas evaluations from the literature do not (Section 3.4.3).
> 4. **"...remains unclear how much of the reported “40% performance gap recovery” actually translates into genuine security threats."** We believe that our evaluation is more desirable than evaluations from the literature, as per the above. However, we agree that how well it captures true potential for harm is an open question. We are excited about future work that would pursue this direction.
>
> Given the evidence that gains reflect real capability transfer rather than style matching or biases of the evaluator models, would you be willing to reconsider your score?

---

### Official Review · Reviewer_g5qm · 2025-10-22

**Soundness:** 2
**Presentation:** 2
**Contribution:** 3
**Rating:** 2
**Confidence:** 4

**Summary:**

The paper introduces elicitation attacks, which aim to elicit harmful capabilities from open-source models. This is achieved by fine-tuning open-source models on benign but adjacent outputs from safeguarded frontier models. Using dangerous chemical synthesis and processing as a case study, the authors show that these elicitation attacks can recover up to ~40% of the performance gap between the original open-source model and the unsafeguarded frontier system.

**Strengths:**

1. This work explores a new and interesting security risk, whether open-source models can pick up harmful skills.
2. The attack method is simple but effective.
3. The results are consistent across different open-source models and measuring metrics.

**Weaknesses:**

1. Lack of mechanistic insight: The paper doesn’t explain why fine-tuning on benign, adjacent chemical tasks enables a weak model to gain harmful capabilities.
2. Limited scalability to new and more complex harmful knowledge: The approach relies on fixed, manually designed benign queries, while frontier models will continue to advance and acquire more sophisticated or creative harmful capabilities. However, assuming there is a fixed ground truth for these benign questions, then the benign prompts and outputs remain static, which prevents open-source models from learning genuinely new or more complex harmful knowledge.
3. Style learning or technical learning: Based on the above two weaknesses, it remains unclear why the observed improvement occurs. I suspect that the improved performance may reflect imitation of the frontier model’s style rather than genuine understanding of harmful chemical synthesis. The evaluation might therefore capture stylistic similarity instead of true technical competence.
4. Poor transferability: The method is only tested on chemical weapon tasks, which rely on strong domain-specific prior knowledge and extensive manual design. It remains unclear whether the same approach would work for other harmful domains, since the paper never defines or automates what qualifies as “adjacent” benign prompts. For example, if an attacker wanted to learn how to build a bomb, what would the corresponding adjacent benign questions be?
5. Unclear evaluation procedure: The evaluation method is difficult to understand and interpret. In the provided example (Figure 1), the judgment that “125–135 °C causes decomposition” seems to rely on the evaluator model’s own reasoning rather than on a reference solution. Which of R1 or R2 represents the reference solution? If m is the structured comparison function, could you clarify how m(W) is calculated, as shown in Figure 1?

**Questions:**

Please refer to the questions in the weaknesses.

---

> ### Author Response · Authors · 2025-11-27
>
> We appreciate your recognition that this work explores a new and interesting security risk with simple but effective methods and consistent results across models and metrics.
>
> Addressing your concerns one by one:
>
> 1. **“Lack of mechanistic insight…”** While mechanistic understanding wasn't our primary goal, we do have some preliminary analyses in the paper:
> - Appendix C shows that models can learn to express previously unexpressed factual knowledge (specifically synthesis routes) and that they make better use of contextual information when synthesis routes are included.
> - Our domain sweep experiments (Section 6) also provide evidence about the mechanism: training on stylistically identical but domain-distant content yields only small gains, showing that performance improvements require domain-adjacent training data, not just style matching.
> - We have also added a discussion of “task overlap” in Appendix J; benign tasks often overlap with subcomponents of the harmful evaluation tasks, even in distinct domains, which could explain why we see uplift overall. We’re happy to include more discussion of this in the paper.
> 2. **“...approach relies on fixed, manually designed benign queries…”** Our approach is not limited to fixed queries. The pipeline we develop in Section 4.3 for the constitutional classifier system and domain sweep experiments uses frontier models to automatically generate queries. In addition, the prompts used for the pipeline developed in Section 4.2 are not truly fixed prompts either. The generation process encourages the inclusion of extra details or constraints to make a question harder or more interesting, even for the same chemical. Stronger frontier models can (and in practice, do) include these extra details and constraints.
> 3. **“...improved performance may reflect imitation of the frontier model’s style rather than genuine understanding of harmful chemical synthesis…”** The domain sweep experiments in Section 6 directly address the style vs. technical learning question. Training on inorganic chemistry produces responses with identical formatting to our organic chemistry training but shows minimal performance gains. This definitively rules out style imitation as the primary mechanism. See the top-level comment as well, which outlines 4 lines of evidence for why we think our uplift is genuine.
> 4. **“...only tested on chemical weapon tasks…”** As discussed in the top-level comment, we believe chemistry domains are an important test case due to their potential for both LLM capability uplift and real-world harm. For example, many frontier labs such as OpenAI or Anthropic specifically focus on chemical or biological weapons uplift as part of their CBRN mitigation strategy.
> 5. **“...evaluation method is difficult to understand and interpret….”** We've clarified Figure 1 as you requested. We’ve also added several sentences to better indicate how the evaluation works. m(W), as specified in Section 4.1 can either be the percentage of rubric keywords present, or the anchored comparison evaluation score, as defined in Section 3.3. See top-level comment as well.
>
>
> We respectfully believe our domain sweep experiments (Section 6), human expert validation (Section 3.4.1), and mechanistic analyses (Appendix C) provide strong evidence that the observed improvements reflect genuine capability transfer. We also believe we have significantly clarified the anchored comparison evaluation / structured comparison evaluation. Additionally, if you disagree that this evidence is sufficient, we would appreciate understanding what additional evidence would be convincing.

---

### Official Review · Reviewer_R5gC · 2025-11-01

**Soundness:** 4
**Presentation:** 3
**Contribution:** 4
**Rating:** 8
**Confidence:** 3

**Summary:**

This paper proposes a new elicitation attack paradigm, showing that benign content generated by frontier models can be used to finetune de-safeguarded open source models, which then exhibit harmful domain performance uplift. The authors validate this on chemical synthesis and processing tasks.

The attack first constructs harmless <prompt, output> pairs by (1) prompts derived from benign compounds in “Compounds” database and then frontier models generate answers, or (2) using relevant topic-based prompting to frontier models that bypass the Constitutional Classifier. Then these harmless pairs are used to fine-tune the open source abliterated models via QLoRA. To quantify the quality of answers,  the authors employ rubric scores and additionally propose the structured comparison. It uses frontier models to generate references and a separate frontier model as a judge to compare tested outputs against references along weighted subgoals. The reliability of this structured comparison is further evaluated by human evaluation, deliberate error detection, and ground truth rating comparison.

Experiments show harmful uplift across multiple open-source models. Ablations examine task type, frontier capability, data scale, and training-domain adjacency.

**Strengths:**

- The motivation of this paper is clear and the intuition is straightforward.
- The discussions from the angles of harmless-to-harmful generalization and ecosystem attacks (i.e., even if both the input and output of the frontier are filtered, a malicious player can still use benign output to train another unguarded open-source model, circumventing guarding strategies at the “system level”) are very insightful to the community. Thanks to the authors for this interesting work!
- This paper is technically sound with comprehensive and systematic evaluations. Dataset collection is also high-quality and thorough.

**Weaknesses:**

- Limited algorithmic novelty, as the attack applies standard SFT/QLoRA on novel data and the evaluation design is regular. The main contribution is problem formulation, data construction, and evaluation framing, rather than a new attack technique.
- Results are conducted on abliterated open-source models. It remains unclear whether common safeguard techniques would partially persist under this attack.
- The role of the frontier model in this attack paradigm is unclear.
    - There is no discussion on why only using the <prompt, output> pairs generated by frontier models can have the attack success. Specifically, what are the fundamental differences between LibreChem baseline and the method? If using a language model to rewrite the original LibreChem dataset into <prompt, output> template, how is the performance? I suggest that the authors address this during rebuttal, as currently, it’s hard to tell whether APGR gain comes from frontier quality, instructional formatting, task targeting, or all of the above.
    - Similarly, why newer/stronger frontier models can yield larger harmful uplift would help to understand this question.
- The authors acknowledge key limitations of structured comparison, and it does affect the evaluation fairness. In the current setup, it is still unclear how much of APGR comes from true procedural correctness versus format/length/style alignment to the references.
- It is appropriate to avoid publishing any sensitive content; however, the paper could still provide aggregate, non-sensitive information of the testing benchmark (e.g., numbers of samples, high-level category balance, source provenance, etc.) to support claims of coverage and fairness.
- Compared to the breadth of the attack analysis, the paper offers rather limited discussion from the defensive perspective.

**Questions:**

See above weaknesses.

**Details Of Ethics Concerns:**

The paper demonstrates strong safety awareness and deliberately avoids releasing sensitive operational information. Still, given the topic and substantial amount of materials, it would be nice to have an individual ethics/safety review.

---

> ### Author Response · Authors · 2025-11-27
>
> Thank you for the positive assessment and for recognizing our work's clear motivation, insightful framing, and systematic evaluation. We're glad you found the discussion of harmless-to-harmful generalization and ecosystem attacks valuable to the community. We further appreciate your suggestion of turning the LibreChem textbook dataset into a prompt-output pair dataset as an additional baseline to test, as we believe its inclusion has led to a more complete paper.
>
> Regarding your specific concerns:
>
> 1. **“Limited algorithmic novelty”** We agree our main contribution is problem formulation, data construction, and evaluation framing rather than novel training techniques.
> 2. **“...unclear whether common safeguard techniques would partially persist under this attack...”** The attack requires adversaries to have access to an open-source model without safeguards, which abliterated models provide. Testing on safeguarded models wouldn't demonstrate this attack vector since the fine-tuned model would refuse harmful requests.
> 3. **“...role of the frontier model in this attack paradigm is unclear…”** We discuss several baselines in Section 4.1 (training on LibreChem textbooks, training on a dataset generated by weak models), and find that the only setting that provides significant uplift is training on a dataset from the frontier model.
> 4. **“If using a language model to rewrite the original LibreChem dataset into <prompt, output> template, how is the performance?”** Since submission, we have run this experiment for Llama 3.3 70B. Specifically, we turned the LibreChem data into a prompt-output pair dataset, and found minimal uplift when fine-tuning (-1.7% APGR). We have a more extensive write up of this experiment in Appendix M.
> 5. **“...hard to tell whether APGR gain comes from frontier quality, instructional formatting, task targeting, or all of the above.”** We rule out that uplift comes from matching the format of frontier responses in Section 6. If by 'task targeting' you mean selecting benign tasks that share sub-skills with harmful evaluation tasks, we analyze this in the new Appendix J, and find that this at least partially explains uplift. We believe that the results in Section 5.1, where performance increases as the frontier model capability increases, indicate that gains come from frontier model capability.
> 6. **“...why newer/stronger frontier models can yield larger harmful uplift ...”**. While we do not analyze exactly why this is the case, we suspect that it is due to stronger models providing more coherent, consistent, and accurate answers.
>  7. **“...unclear how much of APGR comes from true procedural correctness versus format/length/style alignment to the references…”**  We believe our domain sweep experiments (Section 6) demonstrate the observed uplift cannot be explained by style matching. When we train on stylistically identical but domain-distant content, we see the model learn the style, while uplift remains minimal. We have added a paragraph to Section 6 making this explicit. With regards to length, we also explicitly address this confound by ensuring that all evaluated responses are roughly the same length. We mention this in Section 4.1 and have a more in-depth discussion of what effect response length has on APGR, and also what we do to control for this in Appendix G. Finally, we have clarified the terminology to better reflect that the evaluation does not measure similarity to a reference solution (see top-level comment).
> 8. **“... however, the paper could still provide aggregate, non-sensitive information of the testing benchmark (e.g., numbers of samples, high-level category balance, source provenance, etc.)... “** We have added Table 30 in the appendix, which provides detailed information about our testing benchmark including the exact subgoals we use for anchored comparisons, i.e. a high-level view of the skills required for each of our 8 tasks. We use roughly 15-25 samples per task, but there is significant variance due to our filtering out of responses outside the desired length range.
> 9. **“...rather limited discussion from the defensive perspective.”** While defense is not our primary focus, we briefly discuss potential mitigations in the conclusion including KYC requirements for model access and potential restrictions on ostensibly benign but dual-use content.

---

### Author Response · Authors · 2025-11-27
**Top level response**

We thank all reviewers for their thoughtful feedback. Reviewer 1 called our work "technically sound with comprehensive and systematic evaluations" with "clear motivation" and an "insightful" framing. Reviewer 3 noted we "identify a less explored but realistic attack pathway." We note that while Reviewer 2 found our evaluation "difficult to understand," Reviewers 1 and 3 found our methodology sound, with Reviewer 1 specifically praising our "systematic evaluations."

We would like to address some of the largest weaknesses that were mentioned by multiple reviewers below.

# Clarity of Evaluation Methodology
We've clarified our evaluation procedure in response to feedback. In the original submission, we used the term “reference solution” to refer to the set of 10 responses to which outputs are compared for our structured comparison evaluation. This term implies that these references are high-quality or perhaps ground truth responses to the task and perhaps that the structured comparison measures similarity to these references. However, their actual purpose is just to serve as a "calibration point" for judge models: a fixed level of response quality to anchor on so that scores from different tested outputs and judge rollouts are comparable. In fact, the exact prompt we use does not even indicate which response is the reference solution and which is the tested output. To aid with understanding, we therefore have renamed “reference solutions” to “anchor responses”, and the evaluation as a whole from “structured comparisons” to “anchored comparisons.” We have also added an explicit sentence clarifying the role of the anchor responses in Section 3.3

# Real Capability vs Style Matching
Multiple reviewers raised concerns about whether our results reflect genuine capability transfer or mere style imitation. We provide several lines of evidence demonstrating real capability transfer:
1. Our domain sweep experiments (Section 6) directly test this hypothesis. When we train on similar but domain-distant content (e.g., inorganic chemistry), where style transfer would apply but not capability transfer, we find reduced performance gains compared to training on organic chemistry responses (where style AND capability would transfer). This shows that style imitation alone cannot explain our results. We have added a paragraph to clarify this in Section 6.
2. Our rubric evaluation, which measures presence of specific technical information, is independent of formatting or style, and yet we still see large uplift in this metric.
3. We would also highlight that we have strong evidence that our anchored comparison evaluation in general captures true response quality rather than formatting:
- Section 3.4.1 shows our anchored comparison agrees 88% of the time with human chemical weapons experts. This agreement rate exceeds the agreement rate for simpler baseline metrics.
- Human experts rated the anchored comparison transcripts as high-quality in and of themselves as well.
- Anchored comparison also beats out other evaluations (rubrics) from the literature in terms of two properties that good evaluations ought to have: reliable  identification of mistakes in procedures (Section 3.4.2), and rating responses that are known to be correct highly (Section 3.4.3).
4. Appendix G discusses the effect that response length has on measured anchored comparison score. We find that while longer responses generally score slightly higher, the effect size is small. In addition, we introduce several measures to encourage responses to be similar lengths across open-source models and their fine-tuned counterparts, and the frontier models.

# Generalization Beyond Chemistry
We agree that our sole focus on chemistry is a limitation of the work. However, chemical weapons are dangerous, and demonstrating this vulnerability exists in even one high-stakes domain establishes that this attack vector poses real risks worth addressing. Indeed, chemical misues risks have been targeted by OpenAI’s and Anthropic’s model safeguards. Future work could examine whether similar vulnerabilities exist in other high-stakes domains. For example, we believe that this attack could be adapted for biological weapons by training on benign biological tasks.

# Updated Results
We've updated numbers in the domain sweep experiment (Section 6), constitutional classifier system results (Section 4.3), and dataset scaling experiment (Section 5.2). These updates reflect increased sample sizes, a more consistent judge model (gemini-2.5-pro-preview-06-05 rather than gemini-2.5-pro-preview-03-25), re-running results with more consistent hardware, and length suffix optimization (Appendix G). This largely kept results the same but led to changes of a few percentage points in different areas. We believe the updated numbers represent a more robust, accurate, and rigorous measurement of the true APGR.

---

> ### Author Response · Authors · 2025-11-27
> **Full changelog for rebuttal submission**
>
> - Rewrote abstract and introduction to better clarify the ecosystem-level risk framing and to use more precise language around "ostensibly harmless" data (reflecting that our results show such data can in fact enable harm)
> - Reorganized related work into clearer subsections
> - Changed terminology from "structured comparison" to "anchored comparison" and "reference solution" to "anchor response" throughout to better reflect evaluation purpose
> - Added explicit sentence in Section 3.3 clarifying that anchor responses need not be correct, and only serve as a calibration point to get judge models to output more consistent scores
> - Improved wording throughout the paper
> - Clarified Figure 1 to show comparison between rubric and anchored comparison evaluations
> - Improved figure captions throughout to be more self-contained and descriptive
> - Updated results for domain sweep (Section 6), constitutional classifier system (Section 4.3), dataset scaling (Section 5.2), and "Does Abliteration Destroy Model Capabilities?" (Appendix H.1) with increased sample sizes, consistent judge model, consistent hardware, and length suffix optimization
> - Added paragraph in Section 6 explicitly framing domain sweep as test of style-transfer hypothesis
> - Added new baseline (LibreChem prompt-output pairs) as footnote in Section 4.2
> - Added Appendix F: Error bar calculation and noise filtering methodology
> - Added Appendix J: Analysis of task overlap between benign training domains and harmful evaluation sub-goals
> - Added Appendix M: LibreChem prompt-output pair baseline experiment
> - Highlighted in Appendix E that human expert trial used longer transcripts than current evaluation, with validation that condensed version maintains expert agreement
> - Removed appendix section discussing over-refusal examples from the constitutional classifier system, as we felt it unnecessary

---

### Meta-Review · Area_Chair_dL4x · 2026-01-06

**Summary:**

The submission "Eliciting Harmful Capabilities by Fine-Tuning on Safeguarded Outputs" shows that uplift on harmful bio-risk tasks can be generated by finetuning open-source models on adjacent-domain, but not malicious finetuning data generated by the authors from safe, frontier LLMs.

**Reviewer Concerns:**

Reviewers raise a few larger concerns, such as
1. **Understanding the exact role of the frontier model.** The reviewers argue that the submission would have been improved by improving the understanding of the mechanism at play here. The authors conduct additional experiments concerning the hypothesis whether the mechanism is based on the 'style' of the generated data (no), and there is some data included in Appendix C, but overall, the question is not really answered, and the submission mainly functions as a case report of this vulnerability.
2. **Generalizability of Findings**. Reviewers are concerned that results may be too domain-specific. To my understanding, all findings are based on the **8** harmful chemistry tasks from the constitutional classifier paper, which are topical, but indeed limited in scope.
3. **Model-based Evaluation**. A limitation of this work is the model-based evaluation, which although argued by the authors anchored to reference solutions and roughly in line when in relationship to human experts, is an incomplete measure of harm.

Reviewers also raise a few smaller issues that I find either not so concerning, or addressed in the author responses such as the task being straightforward algorithmically, which I find ok for a report of a safety flaw.


Finally, personally, I think this work could have been improved by a more direct, experimental comparison to the various decomposition attack papers, which, while not finetuning-based could be argued to rely on a related underlying mechanism. Or, at least, it would be a nice contribution of this submission to clarify whether these are related mechanisms.



--------
Overall though, I do think the submission is an interesting case study of a safety flaw where info can be extracted from frontier models to uplift open-source models in critical domains, and that this case is interesting enough to be discussed at ICLR.

**Reviewer Scores:**

R5gC no change at 8, g5qm could increase to 4 or 6, d887 may increase to 6.

---

### Decision · Program_Chairs · 2026-01-26

Accept (Poster)